# Filtration and respiration of filter-feeding marine invertebrates are linked through allometric power-law functions

Hans Ulrik Riisgård[1,*] and Poul S. Larsen[2]

## ABSTRACT

Filter feeding in marine invertebrates is a secondary adaptation where the filtration rate ($F$) that provides the food energy to cover the respiration ($R$) increases with increasing body dry weight ($W$), and therefore it may be suggested that the exponents in the equations $F=a_1 W^{b_1}$ and $R=a_2 W^{b_2}$ have, during evolution, become near equal, $b_1 \approx b_2$, ensuring that the $F/R$-ratio $= a_1/a_2$ is nearly constant. Based on published data, we verify the hypothesis of equal allometric power-law exponents and test to what degree the $F/R$-ratio may be used to characterize various adaptations to filter feeding. The available $b$-values for very different taxonomic groups of filter feeders (bivalves, ascidians, crustaceans, polychaetes, jellyfish) covering 8 decades support in most cases the hypothesis of $b_1 \approx b_2$. For obligate phytoplankton filter feeders where $b_1 \approx b_2$ the $F/R$-ratio was used to estimate the critical phytoplankton biomass below which the animal would starve. However, if the food-particle retention efficiency is not constant during an animal's ontogeny the $F/R$-ratio may change according to the size range of particles being captured at the specific stage of development.

KEY WORDS: Filtration, Respiration, $F/R$-ratio, Power-law functions, $b$-exponents

## INTRODUCTION

Filter feeding in all marine invertebrates is a secondary adaptation, and the blue mussel, *Mytilus edulis*, is a well-known example (Cannuel et al., 2009; Gosling, 2015). In mussels, the gills, which may have originated as respiratory organs in ancestors like the protobranchiate mollusks (Yonge, 1939; Jørgensen, 1966; Gosling, 2015), have become greatly enlarged W-shaped water-pumping and cilia-based particle-capturing organs, much larger than needed for respiration (Jørgensen, 1990; Riisgård et al., 2015). Further, it has been suggested that the size of the enlarged gills have been evolutionarily developed to the prevailing low phytoplankton concentration in the sea, and that the energetically costly gills have evolved to be small for continuous feeding, i.e. 'minimal scaling' of the filter-pump, rather than discontinuous feeding using even more enlarged gills (Jørgensen, 1990).

[1]Marine Biological Research Centre, Department of Biology, University of Southern Denmark, 5300 Kerteminde, Denmark. [2]DTU Construct, Technical University of Denmark, 2800 Kgs. Lyngby, Denmark.

*Author for correspondence (hur@biology.sdu.dk)

H.U.R., 0000-0002-8188-2951; P.S.L., 0000-0002-7155-5965

In ascidians, two thirds of the body volume is made up of a greatly enlarged pharynx, which as a secondary adaptation has been developed into a feeding organ. The large pharynx is perforated with small slits (stigmata) with ciliary tracts that create a water current that runs from the inhalant siphon, through the pharyngeal chamber and stigmata into the atrium, and finally out through the exhalant siphon. When the water is pumped across the pharynx wall, suspended particles are trapped on a mucous net continuously produced by the endostyle. The endless mucous net, with retained food particles, is rolled into a cord, passed into the esophagus and eaten.

Filter feeding has secondarily evolved independently in several groups within crustaceans, especially among the small forms, and the feeding mechanisms often differ fundamentally from group to group (Riisgård, 2015). One common feature for all crustacean filter feeders is that the filter-feeding process is true sieving, implying that the mesh size of the filter (filtratory setae) determines the size of the captured suspended food particles. Further, marine calanoid copepods, which represent the crustaceans in the present study, have developed sophisticated mechanochemical sensing of individual phytoplankton cells thus improving their grazing impact in the sea.

Other examples of secondary adaptation to filter feeding may be found among polychaetes. Thus, *Sabella penicillus,* which lives in a tube built from suspended mud, has developed a ciliary crown-filament-pump (Riisgård and Ivarsson, 1990; Riisgård and Larsen, 1995), where compound latero-frontal cilia both pump water and capture suspended food particles by means of the catch-up principle (Riisgård et al., 2000).

The two closely related polychaetes *Nereis diversicolor* and *Nereis virens* both live in shallow soft bottoms. The most conspicuous difference between the two otherwise omnivorous polychaetes is the unique ability of *N. diversicolor* to nourish as a facultative filter feeder (Nielsen et al., 1995). Just as a typical obligate filter feeder *N. diversicolor* may meet its metabolic requirements on a diet of phytoplankton. If the phytoplankton concentration is sufficiently high, *N. diversicolor* shifts from surface-deposit feeding to filter feeding (Riisgård, 1991; Vedel et al., 1993). The worm spins a funnel-shaped mucous net and pumps water through it by vigorously undulating body movements. After a period of pumping, the worm moves forward to swallow the net with entrapped food particles. This feeding behaviour is maintained if the phytoplankton concentration is above the 'trigger' level of 1 to 3 µg chlorophyll $a$ l$^{-1}$. There are no conspicuous morphological differences between *N. virens* and *N. diversicolor*, and filter feeding is therefore considered to be a relatively recent secondary adaptation in *N. diversicolor*.

Predatory filter feeding on zooplankton has been adapted by members of the scyphozoans where the common jellyfish *Aurelia aurita* is a well-known example (Riisgård and Larsen, 2010). During the power stroke contraction of the umbrella water is forced out of the bell cavity, and during the recovery stroke the bell diameter increases so that water moves past the bell margin into the subumbrella cavity. Prey entrained within this water are either

sieved through tentacles lining the bell margin or directly encounter the oral arms or subumbrella surface, which are richly provided with nematocysts (nettle cells). Prey captured on the tentacles are removed by the oral arms and passed to the gut (Costello and Colin, 1995).

The secondary adaptation to filter feeding in *M. edulis* must have involved the development of an enlarged gill-filter pump of increased body mass with filtration rate ($F$) that provides the food energy to cover the respiration ($R$) that increases with increasing body mass ($W$). Therefore, it may be hypothesized that the exponents in the equations $F=a_1W^{b1}$ and $R=a_2W^{b2}$ have, during evolution, become near equal, $b_1 \approx b_2$, ensuring that the $F/R$-ratio, which expresses the litres of water filtered per ml of oxygen consumed, remains sufficiently high to ensure adequate food uptake for maintenance and growth, see also Larsen and Riisgård (2022). Thus, all obligate marine filter-feeding invertebrates face the same challenge of growing on a low concentration of food particles, and this suggests the existence of a common trait among these animals.

This trait is given by the value of $F/R$-ratio that relates to food availability at the living site. The adaptation of an animal to filter feeding can be assessed by knowing the minimum food energy uptake (ingestion) needed to cover its maintenance metabolic energy requirement expressed as the respiration ($R$) measured as the amount of oxygen consumed by the starving animal. The ingestion can be expressed as the volume of water the animal pumps through its filter ($F$) times the food-particle concentration, which depends on the phytoplankton concentration. The ratio $F/R$ expresses the litres of water filtered per ml oxygen consumed=l $H_2O$ (ml $O_2$)$^{-1}$. A minimum value of $F/R$=10 l water filtered per ml $O_2$ consumed for a phytoplankton filter-feeding invertebrate has been suggested by Riisgård and Larsen (2000). If $b_1 \approx b_2$ this implies that the $F/R$-ratio versus $W$ must be near constant and independent of body size if the food-particle capture efficiency remains unchanged during the ontogeny. But this is clearly not the case in mussels, where the planktonic veliger larvae are clearing the ambient water for only sufficiently small – 2 to 6 µm – food particles by means of the velum (Riisgård et al., 1980; Sprung, 1984), whereas metamorphosed juvenile and adult mussels use enlarged gills as a feeding organ to pump water and capture food particles larger than 4 µm with 100% efficiency (Møhlenberg and Riisgård, 1978). Therefore, it may be expected that the $F/R$-ratio is higher in veliger larvae than in adult mussels.

The aim of the present study has been to further substantiate the hypothesis of equal allometric power-law exponents for filtration and respiration versus body size of filter-feeding marine invertebrates, and to test to what degree the $F/R$-ratio may be used to characterize various adaptations to filter feeding. For comparison of $b$-exponents to verify the hypothesis we use published data on $F$ and $R$ obtained on *M. edulis* during its development from veliger larva to adult, measured by the same group of researchers, along with data on various other species, each representing a taxonomic group of filter-feeding invertebrates.

## RESULTS

Table 1 shows $b$-exponents and $a$-values in the allometric power-law functions for filtration $F=a_1W^{b1}$ and respiration $R=a_2W^{b2}$ of six filter-feeding species and three stages of *M. edulis*. The $a$-values have been converted to same units. i.e.: $a_1$=l water filtered h$^{-1}$ g$^{-1}$, and $a_2$=ml $O_2$ h$^{-1}$ g$^{-1}$, which allows estimation of the $F/R$-ratio= $a_1/a_2$=litres of water filtered per ml $O_2$ consumed, provided $b_1 \approx b_2$.

Fig. 1 shows the ranges covered by each of the experimentally determined correlations for $F$ and $R$, over a total range of 8 decades of dry weighs ($W$) of filter-feeding invertebrates. The slopes of $b_1$ and $b_2$ are similar within the individual species, and within the three

ontogenetic stages of *M. edulis*, as suggested by the present hypothesis of $b_1 \approx b_2$.

Fig. 2 shows the $F/R$-ratio of all correlations, where a horizontal slope indicates $b_1 \approx b_2$ and the magnitude of the $F/R$-ratio indicates the degree of evolutionary adaptation to the living site in the sea.

From Table 1 it appears that for *M. edulis* the $b$-exponents for both $F$ and $R$ versus $W$ tend to be higher in veliger larvae and small juveniles $b_1 \approx b_2 \approx 0.9$, falling to $b_1 \approx b_2 \approx 0.66$ in adult mussels. This supports the theory of similar $b$-exponents for filtration and respiration. Further, the $F/R$-ratio ($a_1/a_2$) is high in veliger larvae and small juveniles 71.0 to 79.4 l $H_2O$ (ml $O_2$)$^{-1}$ falling to 16.7 in adults. This reflects that veliger larvae and small juveniles must clear a relatively larger volume of water to capture enough small food particles to cover the metabolism than the adult mussels with large gills, which also capture food particles >4 µm with 100% efficiency.

It is seen from Table 1 that for the obligate filter-feeding ascidian, *Ciona intestinalis*, the $b$-exponents for $F$ and $R$ versus $W$ are $b_1$=0.68 and $b_2$=0.831, respectively, indicating $b_1 \approx b_2 \approx 0.7$ to 0.8. The $F/R$-ratio=13.8 l $H_2O$ (ml $O_2$)$^{-1}$ is comparable to adult *M. edulis*, in agreement with a similar adaptation to mainly feed on phytoplankton.

The $b$-exponents reported for calanoid copepods are $b_1$=0.84 and $b_2$=0.78 thus indicating $b_1 \approx b_2 \approx 0.8$, while $F/R$=37.2 liters $H_2O$ (ml $O_2$)$^{-1}$.

The $b$-exponents for the facultatively filter-feeding polychaete *N. diversicolor* have been reported to $b_1$=1.0 and $b_2$=1.2 thus indicating $b_1 \approx b_2 \approx 1$, while $F/R$=6.8 l $H_2O$ (ml $O_2$)$^{-1}$.

In case of the obligate filter-feeding polychaete *Sabella penicillus*, the $b$-exponents are clearly different, $b_1$=0.24 and $b_2$=0.66.

Finally, for the carnivore jellyfish *Aurelia aurita* $b_1$=0.78 and $b_2$=0.86 indicating $b_1 \approx b_2 \approx 0.8$ to 0.9, while $F/R$=359 liters $H_2O$ (ml $O_2$)$^{-1}$.

Our null hypothesis is $b_1/b_2$=1 and from Table 1 it appears that the mean of $b_1/b_2$=0.96±0.13, which is close to 1.

## DISCUSSION
### Significance of *b*-exponents
The examples of $b$-values for very different taxonomic groups of filter-feeding marine invertebrates given in Table 1 and Figs. 1 and 2 in most cases (but not all) supports the theory of equal allometric power-law exponents, i.e. the filtration rate and thus the food ingestion rate must necessarily be able to satisfy the metabolic need when the organism grows from small to large. Thus, the exponents have during the evolution become near equal $b_1 \approx b_2$ ensuring a certain $F/R$-ratio, which reflects the adaptation to the concentration and type of food available in the surrounding water.

From Fig. 1 it appears that the slopes of $b_1$ and $b_2$ are similar within the individual species, and within the three ontogenetic stages of *M. edulis* in agreement with the present hypothesis, but the '3/4 power scaling law' (Riisgård, 1998; Glazier, 2005).

It may be questioned how representative the selected data listed in Table 1 are. But both exponents have only been published for a few species, each representing a taxonomic group, both exponents $b_1$ and $b_2$ have been published. Most published data are on bivalves, represented in Table 1 by *M. edulis* for which Hamburger et al. (1983) for adult mussels found $b_2$=0.663. However, more recently for *M. galloprovincialis*, Arranz et al. (2016) and Pérez-Cebrecos et al. (2023) found $b_2$=0.644 and $b_2$=0.716, respectively, in fair agreement with the theory of $b_1 \approx b_2$=0.67 in mussels, and further, the $b$-exponents for respiration in a variety of bivalve species have been reported to be $b_1 \approx 0.7$ (Table 7.3 in Gosling, 2015). Likewise, for about a dozen species of filter-feeding bivalves the exponent for filtration has been found to be $b_1 \approx 0.7$ (Table 1 in Riisgård, 2001). In

Table 1. b-exponents of rates of filtration $b_1$ (in $F=a_1 W^{b1}$) and respiration $b_2$ (in $R=a_2 W^{b2}$) of various species of marine filter-feeding invertebrates

| # | Filter feeder | Species | $a_1$ | $b_1$ | Reference | $a_2$ | $b_2$ | Reference | $a_1/a_2$ | $b_1/b_2$ |
|---|---|---|---|---|---|---|---|---|---|---|
| 1 | Bivalve | *Mytilus edulis* veliger larvae 0.02-1.0 µg | $220 \times 10^{-6}$ | 0.864 | Riisgård et al., 1981 | $3.10 \times 10^{-6}$ | 0.902 | Riisgård et al., 1981 | 71.0 | 0.96 |
| 2 | | *Mytilus edulis*, 0.06-10 mg | $25 \times 10^{-6}$ | 1.03 | Riisgård et al., 1980 | $315 \times 10^{-3}$ | 0.887 | Hamburger et al., 1983 | 79.4 | 1.16 |
| 3 | | *Mytilus edulis*, 0.01-1.36 g | 7.45 | 0.66 | Møhlenberg and Riisgård, 1979 | 0.475 | 0.663 | Hamburger et al., 1983 | 16.7 | 1.00 |
| 4 | Ascidian | *Ciona intestinalis* 0.002-0.2 g | 7.08 | 0.68 | Petersen and Riisgård, 1992 | 0.515 | 0.831 | Shumway, 1978 | 13.8 | 0.82 |
| 5 | Crustaceans | Calanoid copepods 0.08-8 mg | 128.83* | 0.84 | Kiørboe and Hirst, 2014 | 3.467** | 0.78 | Kiørboe and Hirst, 2014 | 37.2 | 1.08 |
| 6 | Polychaetes | *Nereis diversicolor* 30-92 mg | 8.87 | 1.0 | Riisgård, 1991 | 1.306 | 1.2 | Nielsen et al., 1995 | 6.8 | 0.83 |
| 7 | | *Sabella penicillus* 5-150 mg | 13.62 | 0.24 | Riisgård and Ivarsson, 1990 | 0.13 | 0.66 | Riisgård and Ivarsson, 1990 | 104.8 | *** |
| 8 | Jellyfish | *Aurelia aurita* 0.01-1.36 g | 0.163 | 0.78 | Møller and Riisgård, 2007 | $0.454 \times 10^{-3}$ | 0.86 | Frandsen and Riisgård, 1997 | 359 | 0.91 |
| | | | | | | | | | x±s.d. | 0.96±0.13 |

*W*, body dry weight; $a_1$, l water filtered h$^{-1}$ g$^{-1}$; $a_2$, ml O$_2$ h$^{-1}$ g$^{-1}$; * $a_1$, l water filtered h$^{-1}$ mg C$^{-1}$; ** $a_2$, ml O$_2$ h$^{-1}$ mg C$^{-1}$; $a_1/a_2$, *F/R*-ratio, litres of water filtered per ml O$_2$ consumed, provided $b_1$, $b_2$; *** not relevant, see text. conversion 1C (µg)=0.12 W (µg); *** not relevant, see text.

*M. edulis* having flat (lamellibranch) gills it may be expected that the gill surface area (*G*) is directly proportional with the filtration rate (*F*), and further that *G* and *F* are proportional to the square of the shell length (*L*) of the mussel, i.e.: $G \approx L^2$ and $F \approx L^2$ (Jones et al., 1992; Pérez-Cebrecos et al., 2023). Likewise, it may be expected that the body dry weight (*W*) is proportional to $L^3$ or reversed: *L* is proportional to $W^{1/3}$, so that $F \approx (W^{1/3})^2 = W^{2/3} = W^{0.66}$ (Riisgård et al., 2014a).

According to Glazier (2005) the nonlinear, ontogenetic shift in *M. edulis* is a 'Type III metabolic scaling', which, however, is not the result of an interplay between pure physical and geometric constraints of the transport of oxygen as suggested by West et al. (1997) who proposed a '3/4 power law' for allometric scaling of respiration rates. But young and fast-growing stages show higher weight specific respiration rates than older and adult stages, which implies that the *b*-exponents tend to be higher: $b \approx 1$ in small (young) mussels falling to $b \approx 2/3$ in larger (older) stages (Riisgård, 1998). Thus, in mussels with enlarged gills there are no constraints of the transport of oxygen deducing a *b*-exponent of 3/4 as suggested by West et al. (1997). Although analysis of a large amount of data on many species has shown that $R \approx W^{3/4}$ this fit only applies when many organisms covering a large span of sizes are compared (Fenchel, 1987).

Because $b_1 \approx b_2 = 0.66$ in *M. edulis* (cf. Table 1) a simple bioenergetic growth model was presented by Riisgård et al. (2014b, equation 3) where the weight-specific growth rate in % d$^{-1}$ is expressed as: $\mu = aW^b$, where $b = -0.34$ and $a = 0.871 \times C - 0.986$, where *W* is in g dry weight of soft parts and *C* is in µg chl *a* l$^{-1}$. Within the concentration range of chl *a* during the productive season in the Great Belt, Denmark, the actual specific growth rates were generally in good agreement with the model. Because $b_1 \approx b_2$ in other filter-feeding invertebrates similar bioenergetic growth models may be developed for these species, as recently done for *Aurelia* sp. where $b_1 \approx b_2 = 0.8$ (Riisgård and Larsen, 2022b). According to this bioenergetic model for the weight-specific growth rate of *Aurelia aurita*, fed brine shrimp, *Artemia salina*, the specific growth will remain high and constant at prey concentrations >6 *Artemia* l$^{-1}$. This statement was verified by Riisgård (2022) who conducted controlled feeding and growth experiments on small jellyfish in Kreisel tanks. It was found that prey organisms offered in concentrations of 25, 50, and 100 *Artemia* l$^{-1}$ resulted in specific growth rates in fair agreement with the model-predicted rates.

### Significance of *F/R*-ratio

To appreciate the significance of the *F/R*-ratio regarding normal functioning (living conditions) we consider three species. First, for *M. edulis* (Table 1) the *F/R*-ratio of 16.7 l H$_2$O (ml O$_2$)$^{-1}$ may be used to calculate the minimum concentration of chlorophyll *a* ($C_{ch}$, chl *a*, µg l$^{-1}$) that must be in the sea to prevent starvation. The following conversion factors are used here: 1 ml O$_2$ corresponds to 0.46 mg C (Stuart and Klumpp, 1984); 1 µg chl *a* corresponds to 40 µg C (Jakobsen and Markager, 2016). The assimilated part of ingestion equals respiration so assuming that the assimilation efficiency is 80% it is found that $F \times C_{ch} \times 0.8 = R$, or $16.7 \times 0.8 \times C_{ch} = 1$ ml O$_2$ = 0.46 mg C, or $C_{ch} = 0.46/(16.7 \times 0.8) = 0.034$ mg C l$^{-1}$ = (0.034/40 =) 0.85 µg chl *a* l$^{-1}$. Therefore, the phytoplankton biomass must be above 0.85 µg chl *a* l$^{-1}$ to ensure a positive energy balance of *M. edulis*. The critical phytoplankton biomass below which the mussel closes its valves has been observed to be between 0.5 (Pascoe et al., 2009; Riisgård et al., 2003) to 0.9 chl *a* l$^{-1}$ (Riisgård et al. 2006), which agrees with the estimated critical concentration of about 0.85 chl *a* l$^{-1}$ to prevent the effects of starvation. Thus, during a starvation period

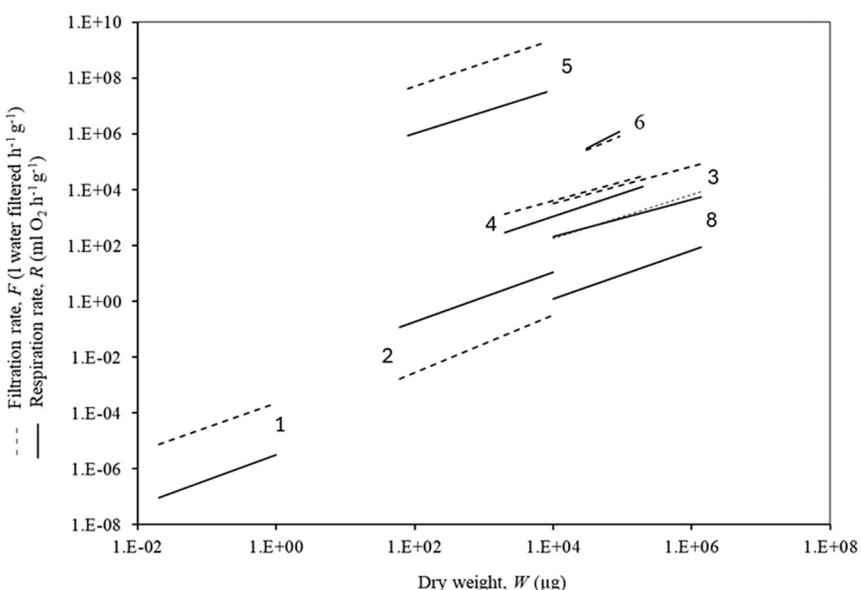

**Fig. 1. Slopes of *b*-exponents for allometric power-law functions of *F* and *R* versus *W*.** 1, mussel *Mytilus edulis* veliger larvae; 2, *M. edulis* post-metamorphic juveniles; 3, *M. edulis* adults; 4, ascidian *Ciona intestinalis*; 5, calanoid copepods; 6, polychaete *Nereis diversicolor*; 8, jellyfish *Aurelia aurita*. Based on data in Table 1 (omitting 7).

*M. edulis* can reduce its respiration rate by reducing its degree of valve opening and thereby the ventilation rate and oxygen uptake (Jørgensen et al., 1986; Tang and Riisgård, 2016), so that the metabolic weight loss is reduced 10 to 12 times (Riisgård and Larsen, 2015).

For the facultatively filter-feeding *N. diversicolor* having $F/R$=6.8 l $H_2O$ (ml $O_2$)$^{-1}$ (Table 1) the minimum chl *a* concentration to prevent starvation is calculated as: $C_{ch}$=0.46/(6.8×0.8)=0.085 mg C l$^{-1}$=(0.085/40=) 2.1 µg chl *a* l$^{-1}$, which is in agreement with the lower trigger level of 1 to 3 µg chl *a* l$^{-1}$ where the polychaete stops filter-feeding and switches to surface-deposit feeding (Vedel et al., 1993).

Next, we consider the obligate filter-feeding polychaete *Sabella penicillus*, where the *b*-exponents are clearly not equal but $b_1$=0.24 and $b_2$=0.66, which is apparently in disagreement with the hypothesis of equal *b*-exponents. However, this may be interpreted as follows. The particle capture mechanism in *S. penicillus* is based on the 'catch-up principle' where compound cilia generate a flow with suspended particles that enter the ciliary region where the same compound cilia during their power stroke catch up with the particles and transfer them to the frontal side of the pinnules to be transported towards the mouth (Riisgård et al., 2000). For particle retention, the lower size limit depends on the spacing between cilia in phase while the upper size depends on the cilia length, which may or may not allow particles to enter the ciliary region to be captured (Riisgård et al., 2000). The same 'catch-up principle' is used by bivalve and gastropod larvae (Riisgård et al., 2000). The particle-size retention spectrum has been measured in *M. edulis* veliger larvae by Riisgård et al. (1980) and Sprung (1984) who found that the clearance rate of 2.5 to 3.5 µm particles was maximum, gradually falling to about 20% of the maximum rate for 1 to 2 µm particles and to 40% for 7.5 µm particles. Sprung (1984) only measured the particle-size retention on veliger larvae of 260 µm shell length. However, the compound cilia (prototrochal cirri) in the gastropod larvae of *Philine aperta*, which also use the 'catch-up principle', increase in length from 10 µm in 130 µm shell-length larvae to 60 µm in 370 shell-length larvae (Hansen, 1991), which indicates that the retention efficiency of larger particles is probably increasing during the growth of the larvae. If a similar increase in length of the compound catch-up cilia takes place in *Sabella penicillus* during its development, this implies an increasing retention efficiency with body size, which may explain why $b_1 < b_2$ because sufficient food ingestion to cover the respiratory need may be ensured by an increased particle-retention of larger particles. So far, however, retention efficiency has only been measured in large adult *S. penicillus*, which shows an optimum retention of about 3 µm particles, with a little lower retention efficiency up to the largest particles of about 8 µm included in the study (Fig. 2B in Jørgensen et al., 1984). Obviously, the $F/R$-ratio in *S. penicillus* should not be estimated as $F/R$=$a_1/a_2$=13.62/0.13=104.8 liters $H_2O$ (ml $O_2$)$^{-1}$ because $b_1 \ll b_2$. But in the case of a large 0.1 g and a smaller 0.05 g *S. penicillus* it is estimated that $F/R$=(7.84/0.028=) 280 and (6.64/0.018=) 369 liters $H_2O$ (ml $O_2$)$^{-1}$, respectively, which may reflect an adaptation to filter feeding on a limited particle-size range in a habitat characterized by a very low phytoplankton concentration.

The very high $F/R$=359 liters $H_2O$ (ml $O_2$)$^{-1}$ estimated for the jellyfish *Aurelia aurita* (Table 1) reflects that it does not feed on phytoplankton but on much lower concentrations of zooplankton being captured by its tentacles.

Finally, the $F/R$-ratio has been estimated for several sponge species. Reiswig (1974) measured *F* and *R* of 3 tropical marine demosponges, *Mycale* sp., *Tethya crypta*, and *Verongia giganta*, with $F/R$-ratios of 22.8, 19.6 and 4.1 liters $H_2O$ (ml $O_2$)$^{-1}$, respectively.

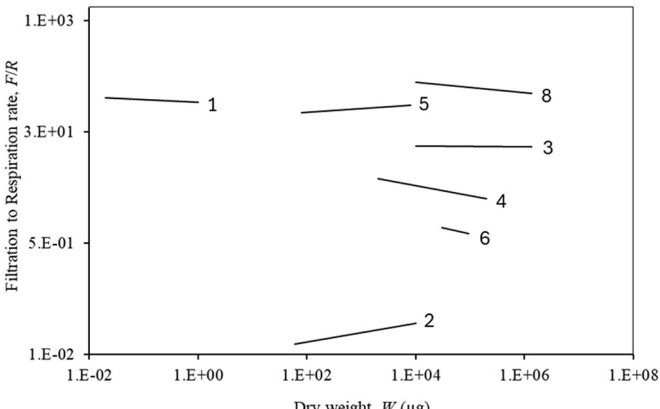

**Fig. 2. *F/R*-ratio versus *W* of filter-feeding invertebrates in Table 1 (omitting 7) (`→).**

The low value for *V. giganta* was suggested to be due to its 'tripartite community'. For the demosponge *Halichondria panicea*, Riisgård et al. (2016) estimated $F/R$=15.5 l $H_2O$ (ml $O_2$)$^{-1}$. Thus, for obligate filter-feeding demosponges with $F/R \approx 20$ liters $H_2O$ (ml $O_2$)$^{-1}$ the minimum chl *a* concentration to prevent starvation is calculated as (cf. above) $C_{ch}$=0.46/(20×0.8)=0.029 mg C l$^{-1}$=(0.029/40=) 0.7 µg chl *a* l$^{-1}$. However, as discussed by Riisgård et al. (2016) the amount of food energy represented by free-living heterotrophic bacteria, cyanobacteria and other small (0.2 to 2 µm) picoplankton, which are also accessible to sponges, may be an important although a somewhat insufficient food source relative to phytoplankton.

## Conclusions

The *b*-exponents in most of the studied species in Table 1 are rather close to being equal, which supports the hypothesis of equal allometric power-law exponents for filtration and respiration in filter-feeding marine invertebrates. But *Sabella penicillus* is an exception that may be explained by increasing retention efficiency of larger food particles due to increasingly longer 'catch-up compound cilia' during the growth of the polychaete so that the ingestion continues to equal the respiration. Thus, a prerequisite for equal *b*-exponents is constant particle-retention efficiency with increasing body weight. The *F/R*-ratio in Table 1 for obligate phytoplankton filter-feeding invertebrates where $b_1 \approx b_2$ this ratio may be used to estimate the critical phytoplankton biomass below which the animal will starve. However, if the food-particle retention efficiency is not constant during the animal's development (ontogeny) the *F/R*-ratio will change according to the range of particle size that are captured at the specific stage of development so that a smaller capture size-range is reflected by a higher *F/R*-ratio.

## MATERIALS AND METHODS
### Definitions used in this study

*R* refers to oxygen uptake, which is the same as oxygen consumption, respiration, metabolism and metabolic energy requirement.

*F* refers to the filtration rate, which is the same as the pumping rate.

*F/R*-ratio: l of water filtered per ml oxygen consumed [l $H_2O$ (ml $O_2$)$^{-1}$].

Clearance rate is the volume of water cleared of particles of a certain size per unit of time. If the particles are retained with 100% efficiency, then clearance rate is the same as filtration rate. Simultaneous clearance of particles of many sizes may be used to lay down the particle-retention spectrum (e.g. Riisgård et al., 1980).

### Criteria for selected data

Oxygen consumption (respiration) does not express the energy cost of filtration in filter-feeding invertebrates. Here, the blue mussel *M. edulis* serves as an example of a filter-feeding marine invertebrate. *M. edulis* closes its valves during starvation to reduce the ventilation rate and thereby save energy by reducing the respiration rate (Riisgård and Larsen, 2015). Thus, when the concentration of algal cells (phytoplankton) becomes very low, *M. edulis* closes its valves resulting in a decline of the filtration rate (also known as the ventilation rate) along with a simultaneous decrease in the oxygen concentration in the mantle cavity and subsequently a decrease in the respiration rate (Riisgård et al., 2003; Riisgård and Larsen, 2015; Tang and Riisgård, 2016). However, subsequent addition of algal cells stimulates the starved mussel to re-open so that maximum filtration rate is soon after restored (Riisgård et al., 2003). The water flow through the mantle cavity and gills of *M. edulis* is laminar and the oxygen uptake is determined by diffusion through boundary layers, and therefore, a reduction in respiration rate is closely correlated with reduced valve gape and reduced filtration (ventilation) rate (Jørgensen et al., 1986). However, this does not reflect physiological regulation of the energetic costs of water pumping but is a consequence of increasing diffusional resistance with decreasing flow (Jørgensen, 1990; Tang and Riisgård, 2016). Therefore, concepts like 'basal

metabolism', 'standard metabolism' and 'metabolism of activity' used in mammalian and fish physiology do not apply to mussels, or other filter-feeding invertebrates. But this has not always been realized, for example, Griffiths and Griffiths (1987) suggested that the metabolic cost of filtration increases logarithmically with filtration rate, going from standard rate to routine rates to end with active rate (see also comments by Tang and Riisgård, 2016).

The oxygen extraction efficiency (*EE*) is defined as the amount of oxygen taken up as related to the total amount of oxygen available in the inhaled flow. In marine filter-feeding invertebrates, where oxygen in the ambient water is taken up by diffusion this implies that only a small fraction of the oxygen dissolved in the water pumped through the animal is available for respiration, and therefore, *EE*=1% or less (Jørgensen et al., 1986; Riisgård and Larsen, 2022a). Reduced filtration rate results in increased *EE*, and therefore, respiration rate is independent of filtration rate above about 20% of water-pumping capacity (Riisgård and Larsen, 2022a). This emphasises the importance of measurement of respiration and filtration rates under similar optimal conditions where the animals exploit their filtration capacity as they are evolutionary adapted to do in nature (but not in the laboratory when unfed, see below).

To obtain comparable data on filtration and respiration rates these parameters must be measured on optimally water-pumping animals. If filtration rate and oxygen uptake (respiration) is measured on an unfed mussel (no algal cells in the ambient water) the mussel will tend to close its valves whereby both filtration and respiration rate become reduced. Therefore, it has been a criterion for selecting data used in the present study that both filtration and respiration rates have been measured on mussels stimulated by algal cells to be wide open and actively water pumping. The same criterion has been applied to the other filter-feeding invertebrates reported on in the present study.

The amount of published data on allometric power-law functions for both filtration and respiration in marine invertebrates is very limited apart from data on bivalves (e.g. Gosling, 2015) where, unfortunately, the criterion on available algal cells and optimal valve-opening degree has not always been met, and further frequently suffers from methodological problems with precise measurement of the filtration rate (Riisgård, 2001). Thus, to avoid possible inter- and intra-specific interpretation problems, only the blue mussel, which has been studied under optimal conditions by the same group of researchers, was selected in the present study to represent the taxonomic group of filter-feeding marine bivalves.

Confounding factors such as temperature and salinity were identical or comparable in studies where both filtration and respiration rates were measured in the same species and therefore not believed to have influenced the scaling *b*-values. Likewise, no energy costs of growth have influenced the measured respiration rates because the animals were not continuously fed algal cells (or fed prey in the case of jellyfish) for a longer period prior to the experiments, and therefore it is the maintenance metabolism that has been measured. In the case of mussels, veliger larvae, juveniles and adults have been evaluated separately because such rare ontogenetic data exist, but in case of other filter-feeder species, data on small (juvenile) and larger (adult) individuals are mixed, which may potentially have influenced the *b*-values. Otherwise, all measurements have been pursued tightly controlled at standardized optimal laboratory conditions.

The investigated filter feeders in Table 1 do not include sponges because they cannot be directly compared to invertebrates with organs, nerves, muscles etc. Nevertheless, our null hypothesis $b_1=b_2$ or $b_1/b_2$=1 agrees with experimental findings for sponges (Reiswig, 1974; Thomassen and Riisgård, 1995; Kumala et al., 2023). Sponges are modular organisms that consist of a set of repetitive aquiferous units or modules with one water-exit osculum per module (Kealy et al., 2019; Kumala et al., 2023). Once a mature module has been formed, its values *F* and *R* no longer change over time. Therefore, when a sponge grows to become a large 'population of modules', both total filtration and respiration rates of the sponge increase linearly with the increasing number of mature modules, i.e. $b_1 \approx b_2 \approx 1$ (Reiswig, 1974; Larsen and Riisgård, 2022). However, the growth characteristics of a small single-osculum sponge explant (module) are fundamentally different from the growth of a large multi-oscula sponge. Nevertheless, lack of further tissue differentiation in the explant (only more of the same when growing,

namely increasingly longer canals separated by walls made up of choanocyte chambers embedded in mesenchyme) also in this case results in $b_1{\approx}b_2{\approx}1$ (Riisgård et al., 2024).

### Mytilus edulis

The blue mussel, *M. edulis*, releases eggs into the seawater where they are fertilized externally, and the embryo develops into a free-swimming larva with a ciliated velum that functions for both swimming and feeding (Widdows, 1991). Water currents for feeding are produced by long compound cilia on the velar edge and suspended food particles are captured by these cilia, which in their power stroke catch up with the particles and transfer them to an oral band to be subsequently carried to the mouth (Riisgård et al., 2000). The larval stage lasts for several weeks before the developed pediveliger settles and undergoes metamorphosis to become a juvenile mussel using gills for water pumping and particle capture (Widdows, 1991). *M. edulis* have been reared in the laboratory from fertilized egg to adult, increasing their body weight by a factor of $10^8$. During this development, *F* and *R* versus *W* in different ontogenetic stages has been measured in several studies.

### Veliger larvae

The clearance (*F*, filtration rate, $\mu l \, h^{-1}$) and respiration (*R*, $nl \, O_2 \, h^{-1}$) of optimally captured 3.5 μm diameter algal cells as a function of size (*W*, dry weight of body tissue) in *M. edulis* veliger larvae between 0.02 and 1 μg were measured by Riisgård et al. (1981) to be: $F=220W^{0.846}$ ($R^2=0.98$) and $R=3.10W^{0.902}$ ($R^2=0.73$), respectively. All data were obtained at 12°C and 27 psu. The measured particle retention spectrum for veliger larvae showed that the clearance of 3.5 μm particles is high and gradually fall to 20% for 1 μm particles and to about 30% of the maximal for 7 μm particles (Fig. 3 in Riisgård et al., 1981).

### Post-metamorphic juveniles

The clearance rate of 100% efficiently captured algal cells >4 μm (*F*, $ml \, h^{-1}$) and respiration rate (*R*, $\mu l \, O_2 \, h^{-1}$) as a function of (*W*, dry weight of tissue) in *M. edulis* young metamorphosed mussels between 60 μg and 10 mg were measured to: $F=0.025W^{1.09}$ ($R^2=0.99$) by Riisgård et al. (1980) and to: $R=315W^{0.887}$ ($R^2=0.97$) by Riisgård et al. (1980). All data were obtained at 12°C and 27 psu.

### Adults

The filtration rate (*F*, $l \, h^{-1}$) and respiration rate (*R*, $\mu l \, O_2 \, h^{-1}$) as a function of (*W*, dry weight of tissue) in adult mussels between 0.011 and 1.361 g were measured to $F=7.45W^{0.66}$ ($R^2=0.99$) by Møhlenberg and Riisgård (1979) and to $R=475W^{0.663}$ ($R^2=0.97$) by Hamburger et al. (1983). All data obtained were at 10 to 13°C and 30 psu.

### Other filter-feeding invertebrates

For comparison with *M. edulis*, which is an obligate phytoplankton filter feeder, other well-studied marine filter-feeding invertebrates have been selected for testing the hypothesis of $b_1{\approx}b_2$ and *F/R*-ratio to reflect these animals adaptation to filter feeding, namely an obligate filter-feeding ascidian (*Ciona intestinalis*), holoplanktonic filter-feeding calanoid copepods, a facultatively filter-feeding polychaete (*N. diversicolor*), an obligate filter-feeding polychaete (*Sabella pinicillus*), and a carnivorous filter-feeding jellyfish (*Aurelia aurita*).

### Ciona intestinalis

The filtration rate (*F*, $ml \, min^{-1}$) as a function of (*W*, g dry weight) in ascidians between 0.002 and 0.2 g was measured to $F=118W^{0.68}$ ($R^2=0.81$) by Petersen and Riisgård (1992). Data were obtained at 15°C and 15-18 psu. The respiration rate (*R*, $ml \, O_2 \, h^{-1}$) as a function of size (*W*, g dry weight) in *C. intestinalis* between 0.01 and 0.8 g was measured to: $R=0.515W^{0.831}$ ($R^2=0.923$) by Shumway (1978). Data were obtained at 10°C and 32 psu.

### Calanoid copepods

The clearance rate (*F*, $l \, h^{-1}$) and respiration rate (*R*, $ml \, O_2 \, h^{-1}$) as a function of size (*W*, mg C) in calanoid copepods were measured to: $F=128.83W^{0.84}$ ($R^2=0.13$) and to: $R=3.467W^{0.78}$ ($R^2=0.48$) by Kiørboe and Hirst (2014). Data converted to reference temperature of 15°C and 34 psu. Conversion of

copepod carbon (C) to body dry weight (*W*) was done according to Halfter et al. (2021, equation 7) as: 1C (μg)=0.12*W* (μg).

### Nereis diversicolor

The filtration rate (*F*, $\mu l \, s^{-1}$) as a function of size (*W*, mg body dry weight) in polychaetes between 4 and 63 mg was measured to $F=2.46W$ ($R^2=0.850$) by Riisgård (1991). The respiration rate (*R*, $\mu l \, O_2 \, h^{-1}$) versus size (*W*, g body dry weight) was measured to: $R=1306W^{1.2}$ ($R^2=0.99$) by Nielsen et al. (1995). All data were obtained at 12-15°C and 18-22 psu.

### Sabella penicillus

The clearance rate (*F*, $l \, h^{-1}$) as a function of size (*W*, g body dry weight) in polychaetes between about 5 and 100 mg was measured to $F=13.62W^{0.24}$ ($R^2=0.985$) by Riisgård and Ivarsson (1990) who also measured the respiration rate (*R*, $ml \, O_2 \, h^{-1}$) versus size (*W*, g body dry weight) to: $R=0.13W^{0.66}$ ($R^2=0.985$). All data were obtained at 17°C and 31-33 psu.

### Aurelia aurita

The clearance rate of jellyfish offered *Artemia* as prey (*F*, $l \, d^{-1}$) versus body size (*W*, mg dry weight) was found to: $F=3.9W^{0.78}$ whereas the respiration rate (*R*, $\mu l \, O_2 \, d^{-1}$) versus size (*W*, mg dry weight) was measured in jellyfish of sizes between 0.05 and 2000 mg to: $R=10.89W^{0.86}$ ($R^2=0.91$) by Frandsen and Riisgård (1997). All data were obtained at 15°C and 15-22 psu.

### Acknowledgements

We thank Professor Douglas S. Glazier and two anonymous reviewers for inspiring comments on the manuscript.

### Competing interests

The authors declare no competing or financial interests.

### Author contributions

Conceptualization: H.U.R.; Formal analysis: H.U.R.; Writing – original draft: H.U.R., P.S.L.

### Funding

This research received no external funding. Open Access funding provided by University of Southern Denmark (SDU) and by Technical University of Denmark (DTU). Deposited in PMC for immediate release.

### Data and resource availability

Data are available upon request to the corresponding author.

### Ethics approval

This is a retrospective study conducted on already published data on marine invertebrates for which formal consent is not needed.

### Peer review history

The peer review history is available online at https://bio.biologists.org/lookup/doi/10.1242/bio.062024.reviewer-comments.pdf

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
