## [Peer Review File · Biology Open]

Filtration and respiration of filter-feeding marine invertebrates are linked through allometric power-law functions - verification of hypothesis

Poul S. Larsen and Hans Ulrik Riisgard

DOI: 10.1242/bio.062024

Editor: Lewis Halsey

Review timeline

Original submission: 16 April 2025

Editorial decision: 22 April 2025

First revision received: 9 May 2025

Accepted: 13 May 2025

Original submission

First decision letter

MS ID#: bio.062024

MS TITLE: Filtration and respiration of filter-feeding marine invertebrates are linked through allometric power-law functions - verification of hypothesis

AUTHORS: Poul S. Larsen; Hans Ulrik Riisgard

I have now reached a decision on the above manuscript.

The reviewer reports are shown at the bottom of this email or can be accessed, together with a copy of this decision letter, by going to:

As you will see, the reviewers raised a number of substantial criticisms that prevent me from accepting the paper at this stage.

They suggest, however, that a revised version might prove acceptable, if you can address their concerns. If you think that you can deal satisfactorily with the criticisms on revision, I would be pleased to see a revised manuscript. We would then return it to the reviewers.

At this stage, we also ask you to ensure your manuscript complies with our formatting guidelines. Provided you are able to fully address the referees' comments, we are positive about publication of your paper (we accept over 95% of revision submissions) and therefore hope you won't mind any extra work involved in reformatting your manuscript at this point.

Please ensure that you clearly highlight all changes made in the revised manuscript. Please avoid using 'Tracked changes' in Word files as these are lost in PDF conversion.

I should be grateful if you would also provide a point-by-point response detailing how you have dealt with the points raised by the reviewers in the 'Response to Reviewers' box. Please attend to all of the reviewers' comments. If you do not agree with any of their criticisms or suggestions please explain clearly why this is so.

Reviewer 1

Title: Filtration and respiration of filter-feeding marine invertebrates are linked through allometric power-law functions - verification of hypothesis

Authors: Riisgard HU, Larsen, PS.

General: In the current study, the authors use published data to test the hypothesis that the allometric scaling factors are the same for filtration and metabolic rate. I enjoyed this study and please excuse if my comments, suggestions, appear ignorant as they come mainly from a background in homeothermic vertebrates. While the scaling is interesting, there appears to be little information on confounding factors such as temperature, salinity, pH etc and how they alter these scaling factors. While ontogeny is discussed, allometric scaling in mammals, for example, tend to "correct" for these by measuring metabolism narrowly defined, i.e. basal. As the historical data used come from within species comparisons "slopes" will depend on conditions for each study, and only the intercepts vary? If so, maybe it would be worth specifying if this is the justification for these comparisons? The reference to the Glazier review highlights the differences between intra- and inter-specific allometry and it may be useful to provide the reader with a short summary of what you actually do here.

Introduction:

-Lines 43-48, I am not sure if there are differences between vertebrates and invertebrates, but later (line 78) it appears what is called respiration is actually O₂ consumption rate. As respiration is something completely distinct, even in fish, maybe change this to O₂ consumption rate.

Data

Lines 78-84: How do you compare (control) for filtering rates if their efficiency (what I think you call retention rate) is only 20-30%? If that varies, then the filtering rate has to increase? The same as if the dissolved O₂ is different, i.e. you have to move more water to extract the O₂? Maybe I am missing something here, but it seems that if you are not tightly controlling your experiment, it will be very difficult to make any comparisons. If I am misunderstanding this, maybe clarify this in the introduction so this becomes more obvious to non-experts? I think this is related to the difference between inter- and intra-specific studies, so clarify this initially will help the reader.

Lines 80-81: I am not sure if there is any journal standard for this, but I would not repeat numbers in the text that are in the tables. For example, all the equations seem to be repeated in the text that is in the tables. Maybe provide the details in the tables and then summarize the trends in the text. The same goes for the other species. Finally, figures are generally easier to interpret than are tables, so maybe include a summary figure?

Line 112, 1306 is 1.306 in the table

Results

Table 1: Would be good to include temperature, and judging from this table, there seems to be no common R among species, which is peculiar and agrees with the Glazier review and also other allometric studies. So the "metabolic" rate does not appear to scale with size? Is this because there is a mixture of adults and juveniles mixed, different temperatures or what could be the reason for this? Apart from the bivalve, are all other species measured as "adults"

Line 131: Interesting conclusion but what are the confidence intervals for these parameter estimates and do they overlap between F and R for the different age classes?

Lines 133-137: This is interesting and what are the "morphological" changes in the filtering/gas exchange organ? Is there a change so that one is controls the rate of water that has to move across the organ? Does this organ change for the same species in habitats with more or less particulates (food) in the water?

Line 138-139: The expressions like " $b_1 \approx b_2 \approx 0.7$ to 0.8 " could be avoided is the authors reported the confidence intervals of the published exponents and then it would also allow the reader to judge if these are really "similar"

Discussion:

-Line 157-162: This is interesting and would be a really neat conclusion if it can be verified when measurements for different species are "standardized". While it may seem surprising, maybe this is logical based on mass-balance. Metabolic rate (in this MS R) changes with size and so does food intake (F in this MS), as they both are related, does this not just state that you need different amounts of food as you grow, which is (if I read this correct), exactly what you propose? Maybe then it does not matter if the external factors, like temperature or dissolved O₂ content, changes, as this will scale for both R and F? Maybe this is worth a discussion

Line 194: Here it becomes confusing when you speak about respiration rates, as this means something different in many vertebrates. So maybe talk about metabolic rate?

Line 191-200: This is very interesting and seems to fit with the literature on homeotherms and maybe provide some comparison other than the West? For example, McNab may have some reference that look at how b varies with age. Also, maybe worth mentioning that most allometric studies in mammals use basal metabolic rate, which does not seem to be a standard used here, i.e. not controlling for age, temperature, etc?

-Lines 238-267: This is an interesting comparison and although this study may be limited, it provides some really neat ideas that could be empirically tested. For example, the clearance retention and could that be low as an animal needs to pass a lot of water past the gills for O₂ uptake?

Reviewer 2

In this paper, the authors test the idea that mass-scaling of rates (here filtration and respiration) are similar (i.e. exponents are identical). The ms is reasonably well written, but could in my opinion be further improved and clarified. The data are compelling, but it is not clear how exhaustive the dataset is compared to all the data that is published on this topic.

Main comments:

visualize the data

refrain from mixing intraspecific (ontogenetic) scaling exponents and interspecific (phylogenetic) scaling exponents

acknowledge that scaling exponents of filtration will depend in part on fluid dynamics

Introduction

The central idea of the study could be better introduced. Is gill size the only way by which filtering invertebrates can modulate oxygen consumption? Surely if the oxygen extraction efficiency changes (e.g. due to lower internal partial pressure, counter current respiratory systems or perfusion of gills) then this assumption of 1 ml O₂ for 10 L of water (line 52) becomes invalid? So I need a bit more background to appreciate this first approximation of the F/R ratio. In addition, the introduction then goes on to challenge one of the necessary assumptions of constant food-particle capture efficiency. So if this assumption is (clearly) violated in the veliger stage, what are the repercussions for the hypothesized constant F/R ratio? I hope these questions help with better framing the central idea of this study

Methods

Are the calanoid copepods (line 106) allometric scaling relationships reflecting multiple species differing in body size? If so, it would be good to see if it is possible to obtain the single species scaling relationships since allometric relationships can differ when considering intra or inter specific relationships.

Results

I was expecting at least one figure to show the collated data from the literature. One figure could be a three panel plot with in panel 1 the relationships for filtration rate as a function of mass, with each line representing a separate dataset, plotted over the size range on which the relationship was calculated. Then in the second panel the same for respiration rate. Then in the third panel the ratio between both, to show that these relationships are more or less horizontal. If you use log-log plots, this should allow one to see all the data for the very different species and life stages.

Line 147: I need some help to see how *A. aurita* is a filter feeder. Note also in the discussion, it is mentioned that it uses its tentacles for feeding (line 269), so I do not see how this is filtering...

Line 149: This is a nice result, and I guess testing this statistically would reveal it to be statistically similar to a value of 1?

In addition, I wonder to what extent variation in F/R (which is greater across species than b_1/b_2) is related to variation in temperature (which is known to impact physiological rates).

Discussion

It is a bit odd to start the discussion with an exposition of sponges just after having mentioned that these animals cannot be compared to the main results presented in the manuscript. My suggestion would be to either also include sponges in the results or remove them from the discussion.

Line 180: Why not include all data and be more exhaustive in the results. Now it seems that there is a single species as an example for each group, but the variation within bivalves only becomes

apparent in the discussion. This is strange. Also, is the scaling exponent by Gossling 2009 for single species, or is it calculated across species (see my comment above about not mixing within and across species exponents). What is the basis for an expectation of an exponent of 2 (line 187)? Is it simply that area is generally squared length or is there something specific about bivalves. Note that a bit more rational may be in order and also that simple geometry may not be appropriate. There is a discussion about this in the fish literature (e.g. Lefevre et al., 2018; <http://dx.doi.org/10.1111/gcb.13978>) and things become more complex when considering mass transfer equations from fluid dynamics (Rubalcaba et al., 2020; <https://doi.org/10.1073/pnas.2003292117>).

Line 191: This paragraph confounds ontogenetic scaling exponents with interspecific scaling exponents. The two types of exponents cannot be compared because the mechanisms are different (plasticity during ontogeny vs evolutionary adaptations).

Line 201: Can a single scaling exponent suffice when in table 1 we see that the scaling exponent for *M. edulis* changes through ontogeny? Or is this because the bulk of growth happens after mussels surpass 10mg, at which their scaling exponent is indeed around 2/3?

Line 230: I would reword this slightly different arguing that the low food concentration results in a cessation of filtering due to low energetic gains and the closing of the valves and cessation of pumping means that energy is not expended and this reduces metabolic rate and hence oxygen demand.

Line 241: I do not follow the explanation of the "catch-up compound cilia" mechanism.

Line 274: I am not familiar with the "tripartite community".

Line 306: In this day and age, data should be made available prior to publication, ideally including the code used to analyse or visualize the data. Note that this can also be made available as embargoed, with a certain date, after which the data become automatically available.

First revision

Author response to reviewers' comments

Reviewer 1: Title: Filtration and respiration of filter-feeding marine invertebrates are linked through allometric power-law functions - verification of hypothesis
 Authors: Riisgard HU, Larsen, PS.

General: In the current study, the authors use published data to test the hypothesis that the allometric scaling factors are the same for filtration and metabolic rate. I enjoyed this study and please excuse if my comments, suggestions, appear ignorant as they come mainly from a background in homeothermic vertebrates. While the scaling is interesting, there appears to be little information on confounding factors such as temperature, salinity, pH etc and how they alter these scaling factors. While ontogeny is discussed, allometric scaling in mammals, for example, tend to "correct" for these by measuring metabolism narrowly defined, i.e. basal. As the historical data used come from within species comparisons "slopes" will depend on conditions for each study, and only the intercepts vary? If so, maybe it would be worth specifying if this is the justification for these comparisons? The reference to the Glazier review highlights the differences between intra- and inter-specific allometry and it may be useful to provide the reader with a short summary of what you actually do here.

Authors' reply: we are pleased that the reviewer "enjoyed this study"; but we also understand the reviewer's problem "with a background in homeothermic vertebrates". Therefore, to make our paper more readable to a wider audience outside a group of experts in filter-feeding marine invertebrates, we have added a rather big volume of new text to the revised manuscript, see added new text at the bottom of this document. Thus, we have extended the Introduction with description of other examples of secondary adaptation to filter feeding and explained the use of *F/R*-ratio in assessing the adaptation to filter feeding, and further, inserted definitions, concepts and criteria for selected data, and explained the ontogeny of mussels, inserted two new figures, and 12 new references.

Introduction:

-Lines 43-48, I am not sure if there are differences between vertebrates and invertebrates, but later (line 78) it appears what is called respiration is actually O₂ consumption rate. As respiration is something completely distinct, even in fish, maybe change this to O₂ consumption rate.

Authors' reply: to avoid any confusion, we now have inserted the following definitions in the revised ms:

R = oxygen uptake = oxygen consumption = respiration = metabolism = metabolic energy requirement.

F = filtration rate = pumping rate.

F/R -ratio: litres of water filtered per ml oxygen consumed = $l \text{ H}_2\text{O} (\text{ml O}_2)^{-1}$.

Clearance rate = volume of water cleared of particles of a certain size per unit of time. If the particles are retained with 100 % efficiency, then clearance rate = filtration rate.

Simultaneous clearance of particles of different sizes may be used to lay down the particle-retention spectrum (e.g., Riisgård et al. 1980).

Data

Lines 78-84: How do you compare (control) for filtering rates if their efficiency (what I think you call retention rate) is only 20-30%? If that varies, then the filtering rate has to increase? The same as if the dissolved O₂ is different, i.e. you have to move more water to extract the O₂? Maybe I am missing something here, but it seems that if you are not tightly controlling your experiment, it will be very difficult to make any comparisons. If I am misunderstanding this, maybe clarify this in the introduction so this becomes more obvious to non-experts? I think this is related to the difference between inter- and intra-specific studies, so clarify this initially will help the reader.

Authors' reply: we understand the reviewer's problem, and therefore, we have inserted a new section "Criteria for selected data" where oxygen uptake and oxygen extraction efficiency are explained along with a mini-review of physiological regulation of filtration rate (ventilation rate) as related to the energetic costs of water pumping. Hopefully, this may help the reviewer to follow our arguments.

Lines 80-81: I am not sure if there is any journal standard for this, but I would not repeat numbers in the text that are in the tables. For example, all the equations seem to be repeated in the text that is in the tables. Maybe provide the details in the tables and then summarize the trends in the text. The same goes for the other species. Finally, figures are generally easier to interpret than are tables, so maybe include a summary figure? Line 112, 1306 is 1.306 in the table.

Authors' reply: we agree that the equations are also given in the text, but here along with supplementary info about R^2 , size range, temperature and salinity. We also agree that figures are easier to interpret, and therefore, we have also inserted two new figures based on data shown in Table 1.

Results

Table 1: Would be good to include temperature, and judging from this table, there seems to be no common R among species, which is peculiar and agrees with the Glazier review and also other allometric studies. So the "metabolic" rate does not appear to scale with size? Is this because there is a mixture of adults and juveniles mixed, different temperatures or what could be the reason for this? Apart from the bivalve, are all other species measured as "adults"?

Authors' reply: temperature and salinity have been added to the text, but not in Table 1 to avoid overloading with data that may not be too relevant for the interpretations made.

Line 131: Interesting conclusion but what are the confidence intervals for these parameter estimates and do they overlap between F and R for the different age classes?

Authors' reply: hopefully, the inserted new Figure 1 may help answering such questions, but we have no confidence intervals.

Lines 133-137: This is interesting and what are the "morphological" changes in the filtering/gas exchange organ? Is there a change so that one controls the rate of water that has to move across the organ? Does this organ change for the same species in habitats with more or less particulates (food) in the water?

Authors' reply: the morphological changes in feeding organs from a ciliated velum in free swimming veliger larvae to gills in adult benthic mussels have been described in the inserted new section (line 74 in original ms).

Line 138-139: The expressions like " $b_1 \approx b_2 \approx 0.7$ to 0.8 " could be avoided if the authors reported the confidence intervals of the published exponents and then it would also allow the reader to judge if these are really "similar"

Authors' reply: we agree, but no confidence intervals are available.

Discussion:

-Line 157-162: This is interesting and would be a really neat conclusion if it can be verified when measurements for different species are "standardized". While it may seem surprising, maybe this is logical based on mass-balance. Metabolic rate (in this MS R) changes with size and so does food intake (F in this MS), as they both are related, does this not just state that you need different amounts of food as you grow, which is (if I read this correct), exactly what you propose? Maybe then it does not matter if the external factors, like temperature or dissolved O_2 content, changes, as this will scale for both R and F ? Maybe this is worth a discussion.

Authors' reply: yes, we propose that more food is needed (F must be increased) when the animal grows (and R therefore increases). We have tried to "standardize" the measurements as explained in the inserted new section (line 72 in the original ms).

Line 194: Here it becomes confusing when you speak about respiration rates, as this means something different in many vertebrates. So maybe talk about metabolic rate?

Authors' reply: the definitions we use for marine invertebrates have now been inserted (line 72 in original manuscript): R = oxygen uptake = oxygen consumption = respiration = metabolism = metabolic energy requirement.

Line 191-200: This is very interesting and seems to fit with the literature on homeotherms and maybe provide some comparison other than the West? For example, McNab may have some reference that look at how b varies with age. Also, maybe worth mentioning that most allometric studies in mammals use basal metabolic rate, which does not seem to be a standard used here, i.e. not controlling for age, temperature, etc?

Authors' reply: "basal metabolic rate" or "basal metabolism", "standard metabolism", or "metabolism of activity" cannot be applied to marine invertebrates (see new section "Criteria for selected data").

-Lines 238-267: This is an interesting comparison and although this study may be limited, it provides some really neat ideas that could be empirically tested. For example, the clearance retention and could that be low as an animal needs to pass a lot of water past the gills for O_2 uptake?

Authors' reply: we are pleased that the reviewer finds that this section provides "really neat ideas" and we agree that more studies are needed. However, as also evident from the F/R -ratios, filter-feeding invertebrates pump much more water than needed for respiration (only about 1 % of the oxygen in the water is extracted).

Reviewer 2: In this paper, the authors test the idea that mass-scaling of rates (here filtration and respiration) are similar (i.e. exponents are identical). The ms is reasonably well written, but could in my opinion be further improved and clarified. The data are compelling, but it is not clear how exhaustive the dataset is compared to all the data that is published on this topic.

Main comments:

visualize the data

refrain from mixing intraspecific (ontogenetic) scaling exponents and interspecific (phylogenetic) scaling exponents

acknowledge that scaling exponents of filtration will depend in part on fluid dynamics

Introduction

The central idea of the study could be better introduced. Is gill size the only way by which filtering invertebrates can modulate oxygen consumption? Surely if the oxygen extraction efficiency changes (e.g. due to lower internal partial pressure, counter current respiratory systems or perfusion of gills) then this assumption of 1 ml O_2 for 10 L of water (line 52) becomes invalid? So I need a bit more background to appreciate this first approximation of the F/R ratio. In addition, the introduction then goes on to challenge one of the necessary assumptions of constant food-particle capture efficiency. So if this assumption is (clearly) violated in the veliger stage, what are the

repercussions for the hypothesized constant F/R ratio? I hope these questions help with better framing the central idea of this study

Authors' reply: to make our paper more readable we have added a rather big volume of new text to the revised manuscript, see added new text at the bottom of this document. Thus, we have extended the Introduction with description of other examples of secondary adaptation to filter feeding and explained the use of F/R -ratio in assessing the adaptation to filter feeding, and further, inserted definitions, concepts and criteria for selected data, and explained the ontogeny of mussels, inserted two new figures, and 12 new references.

Methods

Are the calanoid copepods (line 106) allometric scaling relationships reflecting multiple species differing in body size? If so, it would be good to see if it is possible to obtain the single species scaling relationships since allometric relationships can differ when considering intra or inter specific relationships.

Authors' reply: we agree, but no scaling relationship data are available for different calanoid copepod species.

Results

I was expecting at least one figure to show the collated data from the literature. One figure could be a three panel plot with in panel 1 the relationships for filtration rate as a function of mass, with each line representing a separate dataset, plotted over the size range on which the relationship was calculated. Then in the second panel the same for respiration rate. Then in the third panel the ratio between both, to show that these relationships are more or less horizontal. If you use log-log plots, this should allow one to see all the data for the very different species and life stages.

Authors' reply: we agree and have now inserted two new figures.

Line 147: I need some help to see how *A. aurita* is a filter feeder. Note also in the discussion, it is mentioned that it uses its tentacles for feeding (line 269), so I do not see how this is filtering...

Authors' reply: we understand the reviewer's problem, and therefore, we have extended the Introduction, which is the case of *A. aurita* states (citation):

Predatory filter-feeding on zooplankton has been adapted by members of the scyphozoans where the common jellyfish *Aurelia aurita* is a well-known example (Riisgård & Larsen 2010). During the power stroke contraction of the umbrella water is forced out of the bell cavity, and during the recovery stroke the bell diameter increases so that water moves past the bell margin into the subumbrella cavity. Prey entrained within this water are either sieved through tentacles lining the bell margin or directly encounter the oral arms or subumbrella surface, which are richly provided with nematocysts (nettle cells). Prey captured on the tentacles are removed by the oral arms and passed to the gut (Costello & Colin 1994).

Line 149: This is a nice result, and I guess testing this statistically would reveal it to be statistically similar to a value of 1?

Authors' reply: we are pleased that the reviewer finds this result "nice", but more data (to come in future studies) is needed to make more robust statistical tests.

In addition, I wonder to what extent variation in F/R (which is greater across species than b_1/b_2) is related to variation in temperature (which is known to impact physiological rates).

Authors' reply: in most cases b_1 and b_2 have been made at same temperature and salinity, and therefore, it is not likely that the F/R -ratio has been influenced by these factors.

Discussion

It is a bit odd to start the discussion with an exposition of sponges just after having mentioned that these animals cannot be compared to the main results presented in the manuscript. My suggestion would be to either also include sponges in the results or remove them from the discussion.

Authors' reply: we agree, and this section has now been moved to line 185-198 in the revised ms.

Line 180: Why not include all data and be more exhaustive in the results. Now it seems that there is a single species as an example for each group, but the variation within bivalves only becomes apparent in the discussion. This is strange. Also, is the scaling exponent by Gosling 2009 for single

species, or is it calculated across species (see my comment above about not mixing within and across species exponents). What is the basis for an expectation of an exponent of 2 (line 187)? Is it simply that area is generally squared length or is there something specific about bivalves. Note that a bit more rational may be in order and also that simple geometry may not be appropriate. There is a discussion about this in the fish literature (e.g. Lefevre et al., 2018; <http://dx.doi.org/10.1111/gcb.13978>) and things become more complex when considering mass transfer equations from fluid dynamics (Rubalcaba et al., 2020; <https://doi.org/10.1073/pnas.2003292117>).

Authors' reply: this new text has been added in the new section "Criteria for selected data" (citation):

The amount of published data on allometric power-law functions for both filtration and respiration in marine invertebrates is very limited apart from data on bivalves (e.g., Gosling 2015) where, however, the criterion on available algal cells and optimal valve-opening degree has not always been met, apart from frequent methodological problems with precise measurement of the filtration rate (Riisgård 2001). Thus, to avoid possible inter- and intra-specific interpretation problems only the blue mussel, which has been studied under optimal conditions by the same group of researchers, was selected in the present study to represent the taxonomic group of filter-feeding marine bivalves.

Line 191: This paragraph confounds ontogenetic scaling exponents with interspecific scaling exponents. The two types of exponents cannot be compared because the mechanisms are different (plasticity during ontogeny vs evolutionary adaptations).

Authors' reply: we think that b1 and b2 values for free swimming veliger larvae can be separately evaluated in an evolutionary context but should not be mixed up with b1 and b2 values for adult benthic mussels and therefore evaluated separately in the present study.

Line 201: Can a single scaling exponent suffice when in table 1 we see that the scaling exponent for *M. edulis* changes through ontogeny? Or is this because the bulk of growth happens after mussels surpass 10 mg, at which their scaling exponent is indeed around 2/3?

Authors' reply: we think that separate scaling exponents b1 and b2 apply to three ontogenetic stages (veliger larvae, young post-morphic mussels, and adult mussels. This new text has been inserted in the revised ms (line202) (citation):

The blue mussel, *Mytilus edulis*, releases eggs into the seawater where they are fertilized externally, and the embryo develops into a free-swimming larva with a ciliated velum that functions in both swimming and feeding (Widdows 1991). Water currents for feeding are produced by long compound cilia on the velar edge and suspended food particles are captured by these cilia, which in their power stroke catch up with the particles and transfer them to an oral band to be subsequently carried to the mouth (Riisgård et al. 2000). The larval stage lasts for several weeks before the now developed pediveliger settles and undergoes metamorphosis to become a juvenile mussel using gills for water pumping and particle capture (Widdows 1991).

Line 230: I would reword this slightly different arguing that the low food concentration results in a cessation of filtering due to low energetic gains and the closing of the valves and cessation of pumping means that energy is not expended and this reduces metabolic rate and hence oxygen demand.

Authors' reply: thank you for the suggestion, but we think that the present wording is correct.

Line 241: I do not follow the explanation of the "catch-up compound cilia" mechanism.

Authors' reply: this has been explained a bit more in the new text added to the Introduction (line 58) (citation):

Other examples of secondary adaptation to filter feeding may be found among polychaetes. Thus, *Sabella penicillus*, which lives in a tube build from suspended mud, has developed a ciliary crown-filament-pump (Riisgård & Ivarsson 1990, Riisgård & Larsen 1995) where compound latero-frontal cilia both pump water and capture suspended food particles by means of the catch-up principle (Riisgård et al. 2000).

Line 274: I am not familiar with the "tripartite community".

Authors' reply: it is a kind of symbiosis (as a tripartite relationship in sponges).

Line 306: In this day and age, data should be made available prior to publication, ideally including the code used to analyse or visualize the data. Note that this can also be made available as embargoed, with a certain date, after which the data become automatically available.

Authors' reply: we have not tried to do so before; but perhaps a good idea.

New text inserted into original manuscript

(..... inserted line 42 in original manuscript)

In ascidians two third of the body volume is made up of a greatly enlarged pharynx, which as a secondary adaptation has been developed into a feeding organ. The large pharynx is perforated with small slits (stigmata) with ciliary tracts that create a water current which runs from the inhalant siphon, through the pharyngeal chamber and stigmata into the atrium, and finally out through the exhalant siphon. When the water is pumped across the pharynx wall, suspended particles are trapped on a mucous net continuously produced by the endostyle. The endless mucous net, with retained food particles, is rolled into a cord and passed into the esophagus and eaten.

Filter feeding has secondary evolved independently in several groups within crustaceans, especially among the small forms, and the feeding mechanisms often differ fundamentally from group to group (Riisgård 2015). One common feature for all crustacean filter feeders is that the filter-feeding process is true sieving, implying that the mesh size of the filter (filtratory setae) determines the size of the captured suspended food particles. Further, marine calanoid copepods which represent the crustaceans in the present study, have developed sophisticated mechanochemical sensing of individual phytoplankton cells thus improving their grazing impact in the sea.

Other examples of secondary adaptation to filter feeding may be found among polychaetes. Thus, *Sabella penicillus*, which lives in a tube build from suspended mud, has developed a ciliary crown-filament-pump (Riisgård & Ivarsson 1990, Riisgård & Larsen 1995) where compound latero-frontal cilia both pump water and capture suspended food particles by means of the catch-up principle (Riisgård et al. 2000).

The two closely related polychaetes *Nereis diversicolor* and *N. virens* both live in shallow soft bottoms. The most conspicuous difference between the two otherwise omnivorous polychaetes is the unique ability of *N. diversicolor* to nourish as a facultative filter-feeder (Nielsen et al. 1995). Just as a typical obligate filter-feeder *N. diversicolor* may meet its metabolic requirements on a diet of phytoplankton. If the phytoplankton concentration is sufficiently high *N. diversicolor* shifts from surface deposit-feeding to filter-feeding (Riisgård 1991, Vedel et al. 1993). The worm spins a funnel-shaped mucous net-bag and pumps water through the net by vigorously undulating body movements. After a period of pumping the worm moves forward to swallow the net-bag with entrapped food particles. This feeding behaviour is maintained if the phytoplankton concentration is above the 'trigger' level of 1 to 3 μg chlorophyll *a* l^{-1} . There are no conspicuous morphological differences between *N. virens* and *N. diversicolor*, and filter-feeding is therefore considered to be a relatively recent secondary adaptation in *N. diversicolor*.

Predatory filter-feeding on zooplankton has been adapted by members of the scyphozoans where the common jellyfish *Aurelia aurita* is a well-known example (Riisgård & Larsen 2010). During the power stroke contraction of the umbrella water is forced out of the bell cavity, and during the recovery stroke the bell diameter increases so that water moves past the bell margin into the subumbrella cavity. Prey entrained within this water are either sieved through tentacles lining the bell margin or directly encounter the oral arms or subumbrella surface, which are richly provided with nematocysts (nettle cells). Prey captured on the tentacles are removed by the oral arms and passed to the gut (Costello & Colin 1994).

(.... to inserted line 52 in original manuscript)

The adaptation of an animal to filter-feeding can be assessed by knowing the minimum food energy uptake (ingestion) needed to cover its maintenance metabolic energy requirement expressed as the respiration (*R*) measured as the amount of oxygen consumed by the starving animal. The ingestion can be expressed as the volume of water the animal pumps through its filter (*F*) times the food-particle concentration, which depends on the phytoplankton concentration. The ratio *F/R* expresses the litres of water filtered per ml oxygen consumed = $\text{l H}_2\text{O (ml O}_2\text{)}^{-1}$. A minimum value of *F/R* = 10 $\text{l H}_2\text{O (ml O}_2\text{)}^{-1}$ for a phytoplankton filter-feeding invertebrate has been suggested by Riisgård & Larsen (2000).

(... inserted line 72 in original manuscript)

Definitions, concepts and criteria for selected data

Definitions used in this study

R = oxygen uptake = oxygen consumption = respiration = metabolism = metabolic energy requirement.

F = filtration rate = pumping rate.

F/R -ratio: litres of water filtered per ml oxygen consumed = $l\ H_2O\ (ml\ O_2)^{-1}$.

Clearance rate = volume of water cleared of particles of a certain size per unit of time. If the particles are retained with 100 % efficiency, then clearance rate = filtration rate. Simultaneous clearance of particles of different sizes may be used to lay down the particle-retention spectrum (Riisgård et al. 1980).

Criteria for selected data

Oxygen consumption (respiration) does not express the energy cost of filtration in filter-feeding invertebrates. Here, the blue mussel, *Mytilus edulis*, serves as an example of a filter-feeding marine invertebrate. *M. edulis* closes its valves during starvation to reduce the ventilation rate and thereby save energy by reducing the respiration rate (Riisgård & Larsen 2015). Thus, when the concentration of algal cells (phytoplankton) becomes very low, *M. edulis* closes its valves resulting in a decline of the filtration rate (= ventilation rate) along with a simultaneous decrease in the oxygen concentration in the mantle cavity and subsequently a decrease in the respiration rate (Riisgård et al. 2003, Riisgård & Larsen 2015, Tang & Riisgård 2016). However, subsequent addition of algal cells stimulates the starved mussel to re-open so that maximum filtration rate is soon after restored (Riisgård et al. 2003). The water flow through the mantle cavity and gills of *M. edulis* is laminar and the oxygen uptake is determined by diffusion through boundary layers, and therefore, a reduction in respiration rate is closely correlated with reduced valve gape and reduced filtration (ventilation) rate (Jørgensen et al. 1986). However, this does not reflect physiological regulation of the energetic costs of water pumping but is a consequence of increasing diffusional resistance with decreasing flow (Jørgensen 1990, Tang & Riisgård 2016). Therefore, concepts like ‘basal metabolism’, ‘standard metabolism’ and ‘metabolism of activity’ used in mammalian and fish physiology does not apply for mussels, or other filter-feeding invertebrates. But this has not always been realized. Thus, for example, Griffiths & Griffiths (1987) suggested that the metabolic cost of filtration increases logarithmically with filtration rate, going from “standard rate” to “routine rates” to end with “active rate”, see also comments by Tang & Riisgård 2016).

The oxygen extraction efficiency (EE) is defined as the amount of oxygen taken up as related to the total amount of oxygen passing through the animal. In marine filter-feeding invertebrates where oxygen in the ambient water is taken up by diffusion this implies that only a small fraction of the oxygen dissolved in the water pumped through the animal is available for respiration, and therefore, $EE = 1\%$ or less (Jørgensen et al. 1986, Riisgård & Larsen 2022a). Reduced filtration rate results in increased EE , and therefore, respiration rate is independent of filtration rate above about 20 % of water-pumping capacity (Riisgård & Larsen 2022). This emphasises the importance of measurement of respiration and filtration rates under similar optimal conditions where the animals exploit their filtration capacity as they are evolutionary adapted to do in nature (but not in the laboratory when unfed, see below).

To obtain comparable data on filtration and respiration rates these parameters must be measured on optimally water-pumping animals. If filtration rate and oxygen uptake (respiration) is measured on an unfed mussel (no algal cells in the ambient water) the mussel will tend to close its valves whereby both filtration and respiration rate become reduced. Therefore, it has been a criterion for selecting data used in the present study that both filtration and respiration rates have been measured on mussels stimulated by algal cells to be wide open and actively water pumping. The same criterion has been applied to the other filter-feeding invertebrates reported on in the present study.

The amount of published data on allometric power-law functions for both filtration and respiration in marine invertebrates is very limited apart from data on bivalves (e.g., Gosling 2015) where, however, the criterion on available algal cells and optimal valve-opening degree has not always been met, apart from frequent methodological problems with precise measurement of the filtration rate (Riisgård 2001). Thus, to avoid possible inter- and intra-specific interpretation

problems only the blue mussel, which has been studied under optimal conditions by the same group of researchers, was selected in the present study to represent the taxonomic group of filter-feeding marine bivalves.

Confounding factors such as temperature and salinity were identical or comparable in studies where both filtration and respiration rates were measured in the same species and therefore not believed to have influenced the scaling b -values. Likewise, no energy costs of growth have influenced the measured respiration rates because the animals were not continuously fed algal cells (or fed prey in the case of jellyfish) for a longer period prior to the experiments, and therefore, it is the maintenance metabolism that has been measured. In the case of mussels, veliger larvae, juveniles and adults have been evaluated separately because such rare ontogenetic data exists, but in case of other filter-feeder species, data on small (juvenile) and larger (adult) individuals are mixed, which may potentially have influenced the b -values. Otherwise, all measurements have been pursued tightly controlled at standardized optimal laboratory conditions.

Our null hypothesis is $b_1 = b_2$, but filter-feeding sponges have not been included in Table 1, because they cannot be directly compared to invertebrates with organs, nerves, muscles etc.

(... inserted line 74 in original manuscript)

Mytilus edulis releases eggs into the seawater where they are fertilized externally, and the embryo develops into a free-swimming larva with a ciliated velum that functions in both swimming and feeding (Widdows 1991). Water currents for feeding are produced by long compound cilia on the velar edge and suspended food particles are captured by these cilia, which in their power stroke catch up with the particles and transfer them to an oral band to be subsequently carried to the mouth (Riisgård et al. 2000). The larval stage lasts for several weeks before the now developed pediveliger settles and undergoes metamorphosis to become a juvenile mussel using gills for water pumping and particle capture (Widdows 1991).

New references inserted into revised manuscript

Costello JH, Colin SP (1995) Flow and feeding by swimming scyphomedusae. *Mar Biol* 124:399-406

Griffiths CL, Griffiths RJ (1987) Bivalvia. In: *Animal energetics, bivalvia through reptilia*, Vol. 2 (eds TJ Pandian & FJ Vernberg). pp 1-87. Academic Press, San Diego

Halfter S, Cavan EL, Butterworth P, Swadling KM, Boyd PW (2022) “Sinking dead”—How zooplankton carcasses contribute to particulate organic carbon flux in the subantarctic Southern Ocean. *Limnol Oceanogr* 67:13-25

Jørgensen CB, Møhlenberg F, Sten-Knudsen O (1986) Nature of relation between ventilation and oxygen consumption in filter feeders. *Mar Ecol Prog Ser* 29:73-88

Riisgård HU (2015) Filter-feeding mechanisms in crustaceans (Chapter 15, p. 418-463). In: Thiel, M. & Watling L (Eds) *Lifestyles and feeding biology*, Vol II. *The Natural History of Crustaceans*, Oxford University Press

Riisgård HU, Kittner C, Seerup DF (2003) Regulation of opening state and filtration rate in filter-feeding bivalves (*Cardium edule*, *Mytilus edulis*, *Mya arenaria*) in response to low algal concentration. *J Exp Mar Bio Ecol* 284:105-127

Riisgård HU, Larsen PS (1995) Filter-feeding in marine macroinvertebrates: pump characteristics, modelling and energy cost. *Biol Rev Camb Phil Soc* 70: 67-106

Riisgård HU, Larsen PS (2015) Physiologically regulated valve-closure makes mussels long-term starvation survivors: test of hypothesis. *J Molluscan Stud* 81: 303-307

Riisgård HU, PS Larsen (2010) Particle-capture mechanisms in marine suspension-feeding invertebrates. *Mar Ecol Prog Ser* 418:2

Riisgård HU, Larsen PS (2022) Actual and model-predicted growth of sponges - with a bioenergetic comparison to other filter-feeders. *J Mar Sci Eng* 10: 607. <https://doi.org/10.3390/jmse10050607>

Tang B, Riisgård HU (2016) Physiological regulation of valve-opening degree enables mussels *Mytilus edulis* to overcome starvation periods by reducing the oxygen uptake. *Open J Mar Sci* 6:341-352

Widdows J (1991) Physiological ecology of mussel larvae. *Aquaculture* 94:147- 163

Second decision letter

MS ID#: bio.062024R1

MS TITLE: Filtration and respiration of filter-feeding marine invertebrates are linked through allometric power-law functions - verification of hypothesis

AUTHORS: Poul S. Larsen; Hans Ulrik Riisgard

I am happy to tell you that your manuscript has been accepted for publication in *Biology Open*, pending our standard publication integrity checks. It was accepted on 13 May 2025.

Please follow the suggestion of the Reviewer: 'Line 306: In this day and age, data should be made available prior to publication, ideally including the code used to analyse or visualize the data. Note that this can also be made available as embargoed, with a certain date, after which the data become automatically available'